# Limits of the Process of Rotational Compression of Hollow Stepped Shafts

**DOI:** 10.3390/ma12183049

**Published:** 2019-09-19

**Authors:** Jarosław Bartnicki, Janusz Tomczak, Zbigniew Pater

**Affiliations:** Faculty of Mechanical Engineering, Lublin University of Technology, 36 Nadbystrzycka Str, 20-618 Lublin, Poland; j.tomczak@pollub.pl (J.T.); z.pater@pollub.pl (Z.P.)

**Keywords:** rotational compression, hollow stepped shafts, process limits

## Abstract

This article presents the results of theoretical–experimental testing of the process of rotational compression of hollow stepped shafts. The advantages of the technology and the potential area of its application were discussed. Further on, the limits of the rotational compression technology, preventing the manufacturing of high-quality products, were presented. The research was conducted on the basis of numerical modelling using the finite element method in Simufact Forming, as well as the results of experimental tests performed in a forging machine for rotational compression. On the basis of the results obtained, it can be stated that the process of rotational compression of hollow stepped shafts can be hindered by the following phenomena: uncontrolled slip, deformation of the semi-finished product wall, twisting of the formed steps, material cracking, and deformation of the cross-section of the formed steps. The possibility of those hindrances occurring depends heavily on the assumed technological parameters of the process. For this reason, knowledge of the cause of occurrence of those limitations is vital for the development of the technology and the choice of the process parameters.

## 1. Introduction

The usage of hollow products in machine construction allows one to minimize the manufacturing and operating cost. The lower mass of machines and equipment leads to lower cost of production, and furthermore, allows for lower energy and fuel consumption and exhaust emission during its operation [1]. Moreover, the technical parameters of machines and equipment, in which hollow elements were applied, are enhanced. For this reason, hollow elements are becoming more and more widely used as machine elements [2]. Significant parts of the machine elements that can be hollowed out are axles and shafts. Due to the type of load they carry (mostly bending and torque moments) the strength properties of the hollow products are fairly similar to their traditional equivalents [3,4]. It is, however, only possible to fully exploit the advantages of hollow products if the semi-finished product is formed as hollow in its entirety using plastic forming methods (leaving certain small allowances for finishing). Such an approach requires advanced plastic-forming technology to be developed. This technology would allow the hollow element to be manufactured from both filled and hollow (commercial pipes) billet [5,6,7]. The majority of the currently applied plastic-forming technologies for hollow products and semi-finished products is very complex and requires the use of expensive machines. As a result, an effective application of the currently used plastic forming processes is only possible at the level of large-lot and mass production. Unfortunately, in the case of small-series production and a wide assortment of manufactured products, machining remains the basic manufacturing technology.

For this reason, it is appropriate to continue the research on alternative technological solutions, which would allow effective manufacturing of hollow elements based on plastic forming processes at a small production series. Developing an effective and cheap technology for plastic forming of hollow elements will allow for enhancing the competitiveness of companies, as well as the effectiveness of manufacturing processes. One of the innovative manufacturing methods for hollow machine elements is rotational compression, a process which fits into flexible manufacturing systems. In the rotational compression process, a tube-shaped semi-finished product is compressed by three rolls with the outline, compliant with the geometry of the formed forging [8,9] (Figure 1). The tools are located at every 120° to the axis of rotation of the semi-finished product, that is, sections of a commercial pipe. During the process, the rolls are subjected to rotational movement and, simultaneously, progressive movement in the direction of the material axis. As a result of the rolls’ movement, the material is subjected to rotational movement, during which reduction of the cross-section of each step occurs. In the final phase of the process, progressive motion of tools ceases, whereas rotational movement corrects imperfections of the compressed product.

The major advantages of the proposed technology are high efficiency, simple construction, possibility of forming products from tube-shaped billet, enhancing strength properties of the compressed elements, and easy automation. Due to simple construction of the machine and tools, it is possible to manufacture the products both in small-series and in mass production. Thanks to implementing tube-shaped billets in the process of rotational compression, a significantly lower material use was achieved in comparison to other manufacturing techniques, such as machining or conventional plastic forming methods. Furthermore, in the case of rotational compression in hot conditions, problems with removing scales do not occur, since it falls down under gravity between the nether rolls. Due to the above presented facts, rotational compression technology is very attractive for the industry, and one can assume that it will be developed further. The rotational compression process decreases the production cost by 40%–50%, compared to the traditional methods of metal forming [2,3]. The profitability of this technology decreases the cost of the billet (tube), which is significantly higher than the cost of semi-finished solid products (rods), used in other methods of forming hollow elements. It should, however, be stressed that other factors, apart from the cost of the billet, influence the production cost, including: cost of the tools (tools in the shape of stepped rolls are much cheaper than the tools used in other technologies), insignificant cost of tool regeneration, simple construction of the machines, simple scheme of the process, lower number of operations, and lack of cooling lubricants. All of those factors decrease the production cost. It is also important that the abovementioned factors decrease the cost even in the case of small-lot production.

The research carried out so far shows that rotational compression can be widely used in the case of forming hollow forgings of axles and stepped shafts. It is not, however, limited to shafts with flat steps—worm convolutions [10], teeth [11], and splines [12] can also be obtained. Examples of the elements that can be manufactured are presented in Figure 2. Despite numerous advantages, there are also certain limits to this method, which decrease its applicability and influence the quality of the finished product. For this reason, research on the phenomena hindering the field of application of rotational compression has been conducted. In this study, the most important ones will be presented.

## 2. Subject of Study

The limits to rotational compression are presented in the example of forming a hollow forging of an elementary shaft, where only end steps are reduced (Figure 3). The research was based on numerical modelling using the finite elements method (FEM) as well as experimental testing, performed under laboratory conditions in a special forging machine for rotational compression. The scheme of the process is shown in Figure 4. It was also assumed that the forgings will be formed by rotational compression in hot working conditions from semi-finished products in the form of sections of commercial tubes of similar outer diameter and varied initial wall thickness. As a result, it was possible to generalize the geometrical parameters of the billet to the relative thickness of the wall related to the initial diameter of the billet (*t_o_*/D). Moreover, the kinematic parameters of the process to the relative progressive speed of the tools to their rotational speed (*v/n*) were also generalized.

Numerical simulations of the process were performed in Simufact Forming software for FEM calculations, available on the market. The scheme of the process is presented in Figure 4. The tools, in the form of stepped rolls: 1, 2, and 3, were modelled as rigid objects. The billets—(4 sections of C45 grade steel tubes with the outer diameter *D* = 42.4 mm, wall thickness *t_o_* and length *L_o_* = 120 mm) were modelled as rigid–elastic objects using 8-node elements of the first order.

The material model of C45 grade steel was taken from the data library of the program described in Reference [13] and is described by the following dependency:
(1)$$\sigma_{p}=2859.85\times e^{\left(-0.00312548\times T\right)}\times \varepsilon^{\left(0.000044662\times T-0.101268\right)}\;\;\times e^{\left(\frac{-0.000027256\times T+0.000818308}{\varepsilon }\right)}\times {\dot{\varepsilon }}^{\left(0.000151151\times T-0.00274856\right)},$$
where *T* is the temperature (in the range from 700 °C to 1250 °C), *ε* is the strain, and $\dot{\varepsilon }$ is the strain rate.

It was also assumed that for all the analyzed cases in the initial stage of the process, the entire billet material was heated to 1150 °C, whereas the tools retained a constant temperature of 150 °C throughout the entire process. During the process, the rolls rotated in the same direction at the constant speed *n* = 36 rotations/min and simultaneously moved towards the axis of the billet at the constant speed *v* (depending on the variant of the calculations). The contact surface of the formed material and tools was described by the model of constant friction. Due to the fact that the compression process was conducted in hot working conditions, the friction factor assumed in the calculations was relatively high, *m* = 0.8 [14,15]. Moreover, it was assumed that the heat transfer coefficient for the couple material–tools was equal 20 kW/(m^2^·K), whereas for material–environment, it was 0.35 kW/(m^2^·K). 

In the experimental testing, a forging machine for rotational compression was used. It was developed by the authors of Reference [16]. The machine is comprised of segments and 10 basic systems (Figure 5): carrier frame—1, power unit—2, toothed roll stand—3, forming roll stand—4, drive transmission unit—5, torque measurement system—6, system of drive transmission to the forming rolls—7, hydraulic drive of the forming rolls—8, power supply and control system—9, and measurement system (not marked in the figure). The structure of the machine allows obtaining the kinematics of the tools and the workpiece compliant with the assumptions.

Correct forgings determined by MES and formed during the test from semi-finished products of varied relative wall thicknesses are shown in Figure 6. It can be observed that regardless of the applied research method (numerical and experimental), the proposed process allows forming products in a wide range of wall thicknesses. A characteristic trait of the process is an intensive material flow in the radial direction (in the direction of the axis of the forging), which results in the increase of wall thickness in the areas that come in contact with the tools. Moreover, a significant strain heterogeneity was observed in the surface areas (Figure 6a), caused mostly by circumferential material flow.

An increase in the length of the formed steps is also visible, caused by the material flowing in an axial direction mostly in surface layers. This causes the front surfaces of the formed steps to be concave. It is also important to mention that despite assuming the similar value of reduction ratio for all the analyzed cases, an increase in strain and its elevated heterogeneity can be observed, along with an increase in the relative initial wall thickness (increase of the relative initial wall thickness from 0.095 to 0.26 causes effective strain to increase three times at the same value of reduction ratio of the outer diameter). The obtained FEM results were confirmed (in terms of the shape and dimensions of the forgings) during experimental testing (Figure 6b). The results indicate that the proposed technology allows forming of forgings in the wide range of wall thicknesses. 

In many cases, however, even a slight change of the parameters causes the products to be deformed. Among the basic technological parameters influencing the rotational compression process, and simultaneously the quality of the finished product, there are: initial and final temperature of the material, relative wall thickness (*t_o_/D*), relative progressive speed of the tools (*v/n*), and deformation ratio *(**δ = D/d*) in the area of the formed steps.

## 3. Phenomena Limiting the Realization of the Rotational Compression Process

One of the objectives of the study was to determine the phenomena hindering the rotational compression process. In the majority of cases, the observed limits are directly connected to the assumed process parameters. Among the most important phenomena negatively impacting the rotational compression of tube-shaped billet, there are: uncontrolled slip between the material and the tools, twisting of the cross section of the formed steps, laminating the wall of the billet, longitudinal cracking of the material in the area of the hole, and deformation of the cross section of the formed step and hole.

### 3.1. Uncontrolled Slip of the Tools 

A consequence to the uncontrolled slip of the tools is deformation of the formed step (Figure 7). An uncontrolled slip occurs when the rolling resistance becomes more significant than tangential friction forces on the contact surface of the couple metal, which is the tools (causing the forging to rotate). As a result, the tools cease to rotate the forging (Figure 7b) and instead strip the top layers of the forging. This phenomenon may be caused by numerous factors. Among the most important ones, there is relative wall thickness (*t_o_/D*). An uncontrolled slip is most likely to occur at insignificant wall thicknesses, where the rigidity of the wall on the formed step is insignificant. Another factor is the value of the relative speed of the progressive motion to the rotational movement of the tools *(v/n*). The risk of an uncontrolled slip occurring is much higher for greater values of the progressive motion of the tools, due to high deformation ratio per one rotation. This causes the rolling resistance to increase and hinders the rotation of the forging. Furthermore, the assumed value of the friction factor (related to the state of the tool surface) significantly influences the rotational compression process. Additional lubrication also facilitates the occurrence of slips, due to decreasing friction forces. A similar situation occurs during the process of cold forming, where the values of friction factor are significantly lower than those occurring in the hot forming process.

### 3.2. Twisting of the Formed Step 

This phenomenon is caused by the twisting of the forging due to the difference in peripheral speeds between the subsequent steps of the forging. Twisting of the step leads to strain concentration in the transition area between the steps (Figure 8), which can cause the loss of material cohesion in this area. Such a limit occurs usually during forming of the subsequent steps of the forging with significant differences in deformation ratio (a significant difference of the diameters between the compressed steps). It is, however, to be mentioned that in the majority of cases the forming process is conducted with relatively low values of deformation ratio (*δ* < 1.5) that do not cause the material cohesion loss. Another cause of the phenomenon may be excessive cooling of the material of the formed forging, which causes plasticity loss to occur. The torsion of the cross-section of the end steps of the shaft might be minimized by decreasing friction. In the majority of these technologies, however, the processes are realized without lubrication, which causes the values of friction forces to be very significant. However, reduction of friction forces, e.g., by applying lubrication, increases the danger of the tools slipping, which leads to deformation and crushing of the formed steps (especially at greater values of tool speed and significant cross-section reductions).

### 3.3. Laminating of the Wall of the Semi-Finished Product in the Area of the Formed Steps 

This phenomenon leads to ovalization of the cross-sections of the pivots in the beginning, and later on, to the deformation of the formed steps of the forging (Figure 9). This phenomenon is caused by the insufficient relative radial displacement of the tools in relation to rotational speed of the rolls (*v/n* < 1 mm/rotation). As a result, the material does not flow radially in the direction of the axis of the forging, but is instead laminated and flows in the tangential direction, which causes the length of the wall circumference to increase. This process is quite similar to cross-laminating of a sleeve. In order to eliminate the risk of ovalization of the cross-section as a result of the wall laminating (especially for the process performed in hot working conditions), higher relative speed of the tools ought to be assumed (*v/n* > 1 mm/rotation).

### 3.4. Longitudinal Cracking of the Inner Wall of the Formed Step 

Often, inner cracks occur during the processes of plastic forming of axially symmetric full elements. Those cracks are usually caused by low-cycle material fatigue. In the case of rotational forming of hollow elements, cracks caused by cyclic strain may also occur. In this case, however, the cracks are usually located in the surface layers of the inner wall of the formed steps. It appears that the risk of the material cracking depends on the value of the deformation ratio, the initial thickness of the wall, and the relative speed of the tools. It was observed that the risk of cracking increases along with the decrease of diameter reduction. A similar situation can be observed in the case of the wall thickness, where the risk of the inner wall of the formed step cracking increases along with the increase of the initial thickness of the wall of the semi-finished product. The relative displacement speed of the tools is observed to influence the cracking differently. 

Along with the increase of the speed of tool dislocation in relation to rotational speed, the risk of material cracking decreases. This phenomenon can be explained by the decrease in the number of rotations of the forging. The distributions of the damage criterion according to Cockcroft–Latham, determined during the calculations, for different tool speeds (other parameters remain constant) are shown in Figure 10.

### 3.5. Deformation of the Formed Step

In the case of compressing steps in the elements with less significant relative wall thicknesses (*t_o_/D* < 0.1), a deformation of the cross-section of the formed step may occur (Figure 11). For this reason, forgings from such elements can only be formed with low deformation ratio values (*δ* < 1.2) and relatively low speeds, not exceeding *v/n* < 2 mm/rotation. 

The main reason for the deformation of the cross section is the low rigidity of the semi-finished product’s wall. Due to the pressure of the tools, loss of stability occurs, which results in the deformation of the cross-section in the first phase (triangulation), and then the wall becomes entirely crushed. In relation to the slip, the material is constantly rotated by the rolls at stability loss. In the first phase of the compression process of the thin-walled elements, the cross-section is subjected to significant ovalization, and later on, the wall of the element becomes crushed (Figure 12). On the basis of this information, it was stated that the minimum wall thickness allowing to obtain defect-free forging is *t_o_/D* = 0.094.

The ovalization and subsequent deformation of the compressed steps are indubitably connected to the relative speed of the tools (*v*/*n*) related to the rigidity of the wall of the reduced semi-finished product. For this reason, tests of compression in hot working conditions of thin-walled elements were conducted at lower speeds *v/n* = 0.5 mm/rotation (*t_o_/D* = 0.07 and *v/n* = 0.5 mm/rotation). Decreasing forming speeds allowed one to eliminate the total crushing of the formed steps (lack of slip). However, at these technological parameters, a deformation of the cross-section was observed, resulting in a significant angularity of the formed steps. In this case, the obtained outline of the cross-section was similar to a pentagon (Figure 13). Such a deformation was caused by excessive cooling of the material caused by the length of the process. The temperature drop caused the decrease of the material plasticity. As a result, plastic material flow in the radial direction was hindered. This phenomenon resulted in the laminating of the wall, loss of stability of the material, and deformation of the steps. 

Another limit to the process, observed during the test, was the full closing of the gap of the hole in the formed steps. This phenomenon was observed during the forming of tube elements with greater wall thicknesses (*t_o_/D* > 0.16) and relative speeds (*v/n* > 8 mm/rotation). According to the numerical analyses, the amount of material moved in the radial direction increases, along with the increase of deformation ratio and progressive speed of the tools. At the same time, a decrease in the length of the formed steps occurs. For this reason, a full closing of the gap may occur at higher values of deformation ratio (*δ* ≥ 1.5) and forming speed (*v/n* ≥ 8 mm/rotation). 

This phenomenon is connected to the rapid increase of the resistances of plastic material flow, which causes intensive ovalization of the cross-section and hindering of the rotation of the forging, which causes the tools to slip. This, in turn, results in deformation and crushing of the formed steps (Figure 14).

The risk of the slip as a result of the gap of the inner hole closing can be eliminated by decreasing the forming speed, which can, however, be connected to excessive cooling of the material. For this reason, in such cases it is advised to conduct this process in two stages. In the first stage, the forging is compressed at a significantly lower deformation ratio than required. Then, the progressive motion ceases, the shape of the forging is calibrated and material is evenly distributed, both on the outline and length of the steps. In the second stage, the tools once again form the calibrated element until the required deformation ratio is obtained. It is, however, to be mentioned that the realization of this process according to this scheme may lead to excessive cooling of the material (increasing duration of the process). For this reason, additional heating of the material before the second stage of the process is required. The forgings compressed in two stages with an additional heating of the material in between stages are shown in Figure 15. The shafts were formed from the semi-finished product with relative initial wall thicknesses *t_o_/D* equal, respectively, 0.16 and 0.29 with total deformation ratio *δ* = 1.9, and relative forming speed *v/n* = 8 mm/rotation.

Upon close inspection, a possibility of refractions occurring on the surface of the hole of the formed steps (Figure 16) was detected. In the case of rotational compression, the refractions on the surface of the hole are a sign of stability loss occurring for this area of the forging, which results in plastic buckling of the material. 

This phenomenon is caused by significant values of circumferential compressive stresses that are concentrated during the forming phase in the outer layers, located directly below the tools and in inner areas, located near the hole. The concentration of the circumferential stresses in the outer surfaces is caused by an increase in deformation ratio and ovalization of the cross-section occurring during forming. In this area, however, the material is supported by the rolls, which prevents its deformation. The concentration of compressive stresses in the surface layers of the hole is caused by the material flow in the direction of the axis of the forging. The area of the occurrence of the maximum circumferential stresses in the surface layers of the hole is very extensive and encompass over 65% of the circumference of the hole of the forging, which facilitates the occurrence of refractions on the surface of the hole. Along with increase in the deformation ratio or thickness of the billet wall, the risk of refraction of the hole surface increases. This phenomenon is caused by the necessity of radially displacing significant amounts of material. The refraction of the hole surface can be eliminated or limited by supporting it with a mandrel.

The article does not present an exhaustive list of limits to the process, but presents the ones most likely to occur due to the shape of the products manufactured using this method. The less frequently occurring phenomenon is angularity of the cross-section of the formed steps, which may be caused by the low relative speed of the tools, too low temperature of the billet, and significant billet wall thickness. Another limit to the process is the fact that it is not possible to manufacture an element with a length exceeding five times the length of the billet. This limit concerns the length of the reduced steps. Above this value, stability loss of the wall of the semi-finished products may occur, which results in intensive ovalization of the cross-section and crushing of the element.

## 4. Conclusions

The majority of the currently applied plastic forming processes of hollow elements are rather complex and require complex machines and appliances. This increases the financial expenses required for production launching. This fact makes such technologies cost-effective only in the case of large-lot and mass production. The proposed method can be applied for manufacturing small series of products, as well as mass production due to the usage of simple appliances and tools. Using a tube-shaped billet allows for a significant decrease of material and energy usage, which results in lowered production costs. As a result, the numerical calculations verified with the conducted experimental testing the validity of using the rotational compression method for manufacturing a hollow semi-product using a tube-shaped billet. As a result of the research, the causes for the occurrence of the phenomena interrupting the stability of the compression process were determined, including: uncontrolled slip, deformation of the cross-section of the steps, laminating of the wall, and crushing of the wall. During the testing, the influence of technological and geometrical parameters of the semi-finished products on the risk of the occurrence of those phenomena was also investigated. Based on the numerical calculations and experimental testing, the following conclusions were drawn:
The results of numerical simulations indicate that during the rotary compression process of billet with thicker walls, cracking of the inner wall of the formed steps might occur. The values of the Cockcroft–Latham integral in those areas exceed limit values. No broader knowledge concerning the influence of the complex stress state (characteristic for rotational compression) on limit values of Cockcroft–Latham integral is available.The rotational compression process can be interrupted by the following factors: uncontrolled slip, twisting of the formed steps, laminating of the walls of the steps, deformation of the cross-section, and longitudinal cracking of the wall of the formed steps. The occurrence of these phenomena depends on the assumed process parameters.Rotational compression of forgings with extreme neckings is possible at relative wall thicknesses in the range *t_o_/D* from 0.094 to 0.26 and ensures the defect- and deformation-free finished products.Forming forgings with extreme neckings and insignificant wall thicknesses (*t_o_/D* < 0.094) at the relative speed *v/n* > 1 mm/rotation is difficult to achieve, and the risk of stability loss of the wall and its crushing is high.Decreasing the forming speed (*v/n* < 1 mm/rotation) of the thin-walled semi-finished products (*t_o_/D* < 0.094) allows one to eliminate the deformations of the formed steps. It can, however, cause another deformation of the cross-section in the form of intensive angularity of the cross-section of the formed steps. Such deformation is caused by excessive cooling of the material, connected to the long duration of the process.

## Figures and Tables

**Figure 1 materials-12-03049-f001:**
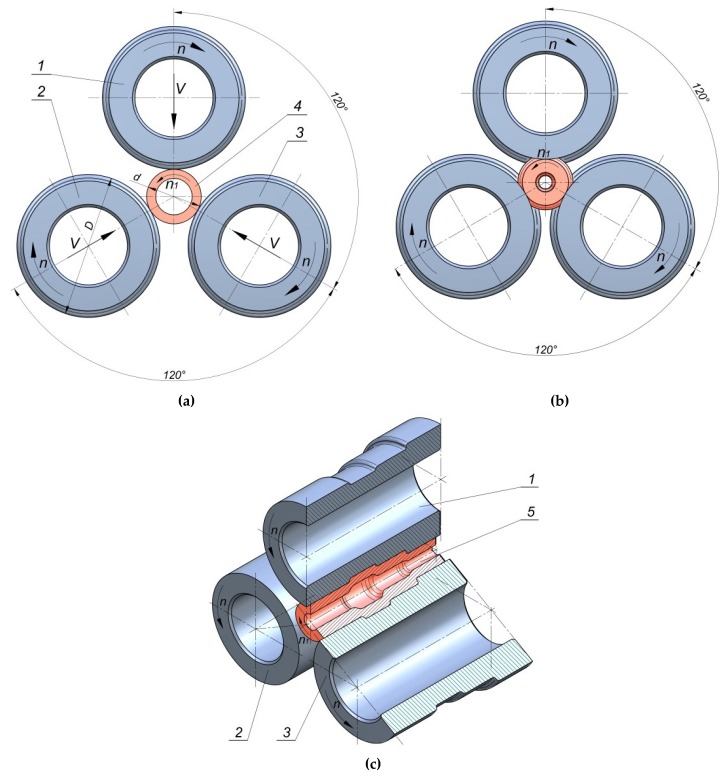
Scheme of rotational compression of hollow elements: (**a**) beginning of the process; (**b**) calibration of the forging; (**c**) end of the process and a formed forging. 1,2,3—rolls, 4—billet, 5—formed forging.

**Figure 2 materials-12-03049-f002:**
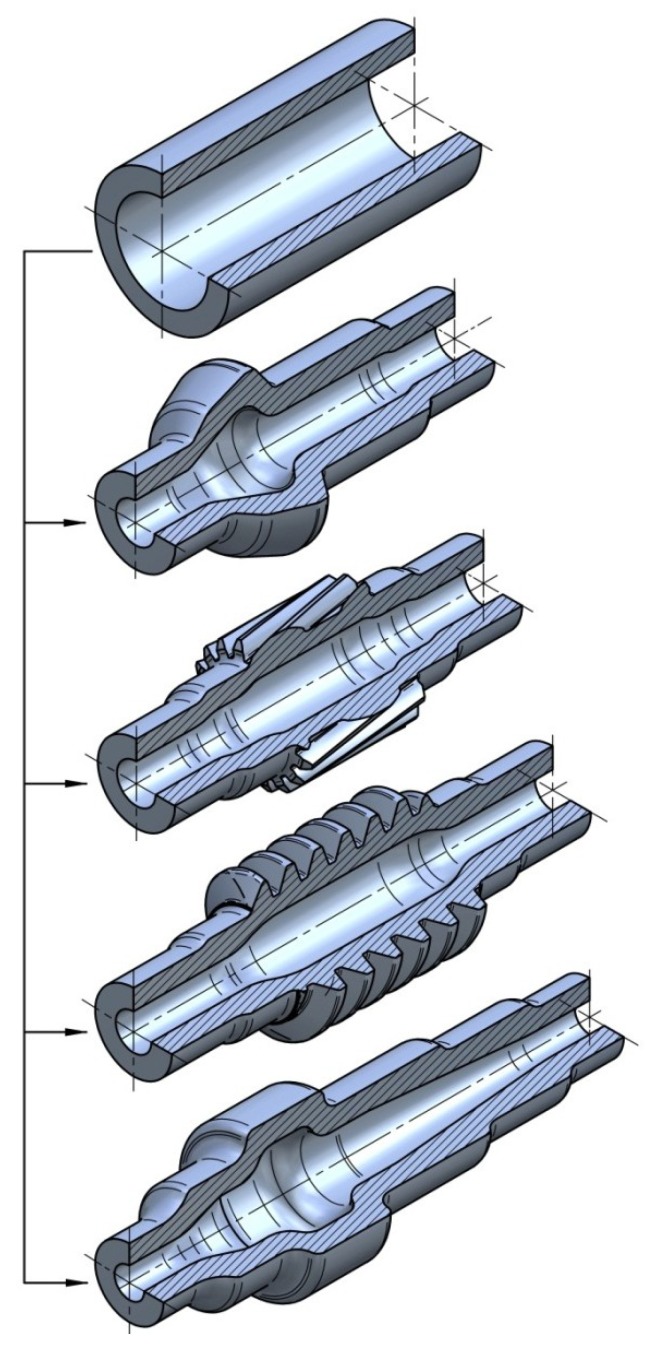
Examples of forgings that can be manufactured by rotational compression.

**Figure 3 materials-12-03049-f003:**
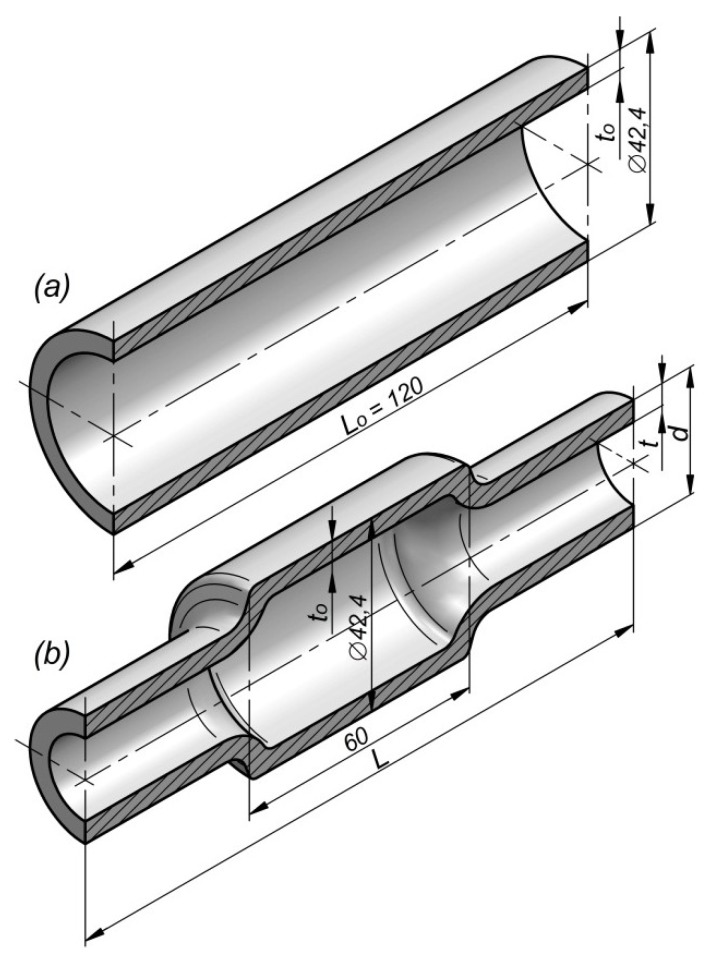
Shapes and dimensions of: (**a**) the compressed semi-finished tube, (**b**) compressed forging of the elementary hollow shaft.

**Figure 4 materials-12-03049-f004:**
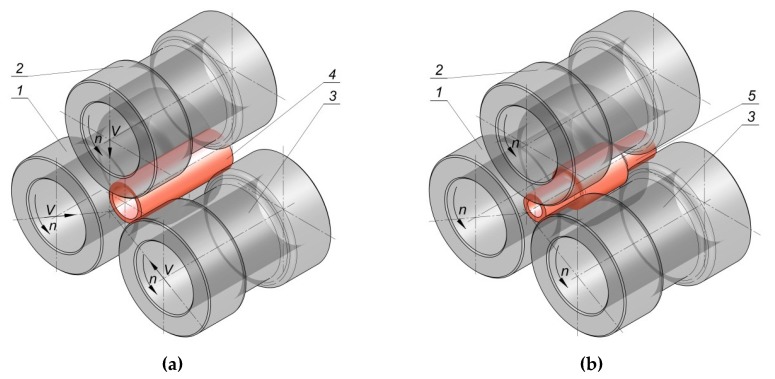
Scheme of rotational compression of a hollow forging of an elementary shaft: (**a**) beginning of the process; (**b**) end of the process, 1,2,3—forming rolls, 4—billet in the form of a section of a tube, 5—finished product.

**Figure 5 materials-12-03049-f005:**
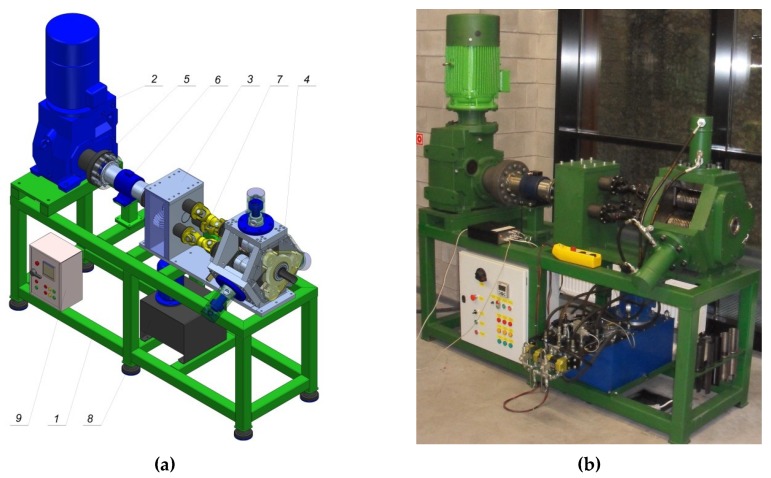
Forging machine for rotational compression: (**a**) 3D model; (**b**) photography of the machine; 1—carrier frame, 2—power unit, 3—toothed roll stand, 4—forming roll stand, 5—flexible coupling, 6—torque measurement system, 7—jointed shafts, 8—hydraulic drive, 9—power box.

**Figure 6 materials-12-03049-f006:**
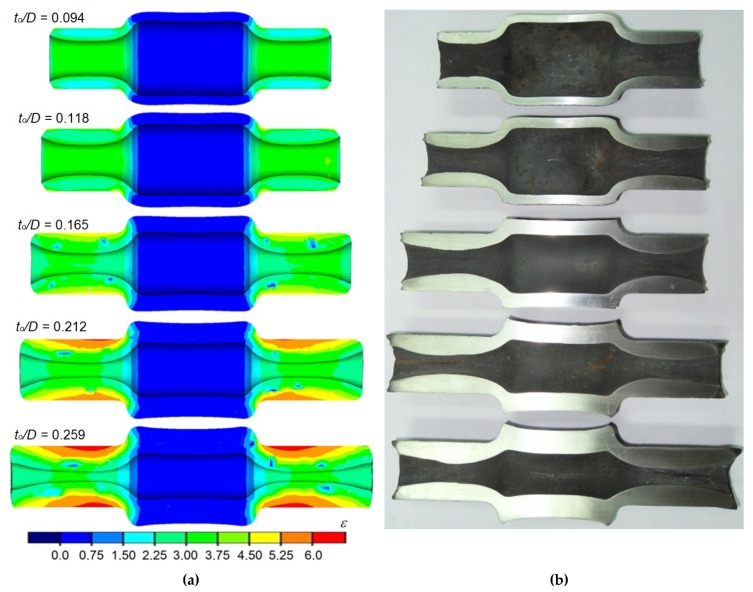
Shape of the formed shafts with extreme neckings: (**a**) finite element method (FEM)-determined (along with the effective strain distribution); (**b**) obtained during laboratory testing (*δ* = 1.5; *v/n* = 4 mm/rotation).

**Figure 7 materials-12-03049-f007:**
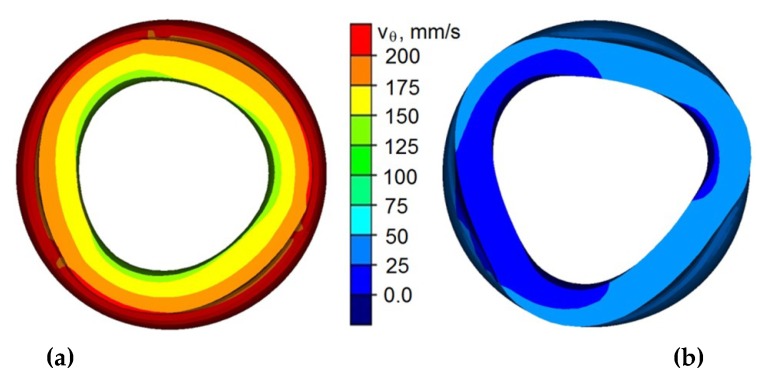
Distribution of the peripheral speed of the formed element: (**a**) in the stable phase of the process; (**b**) during uncontrolled slip.

**Figure 8 materials-12-03049-f008:**
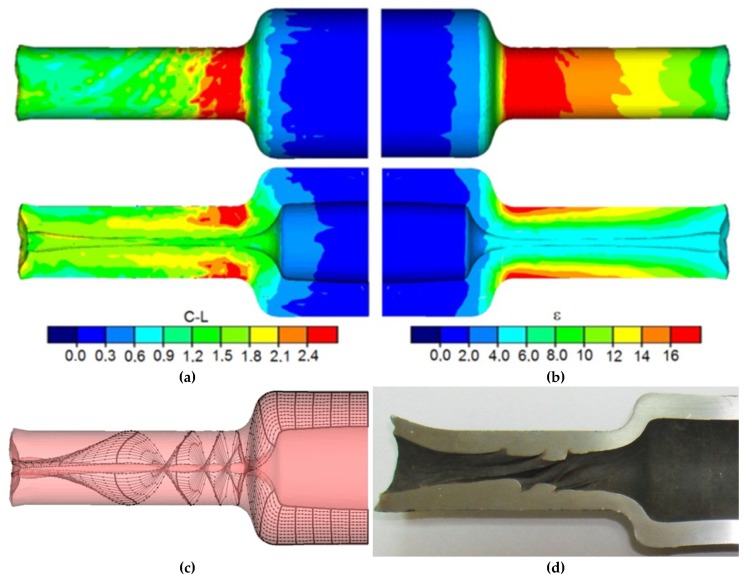
Finite element method (FEM)-determined shape of the end step of the forging, compressed with the parameters: *t_o_/D* = 0.21; *δ* = 2.1; *v/n* = 6 mm/rotation: (**a**) distribution of the damage criterion according to Cockcroft–Latham, (**b**) effective strain distribution, (**c**) twisting of the cross-section of the forging, and (**d**) axial section of the forging of the obtained during laboratory testing.

**Figure 9 materials-12-03049-f009:**
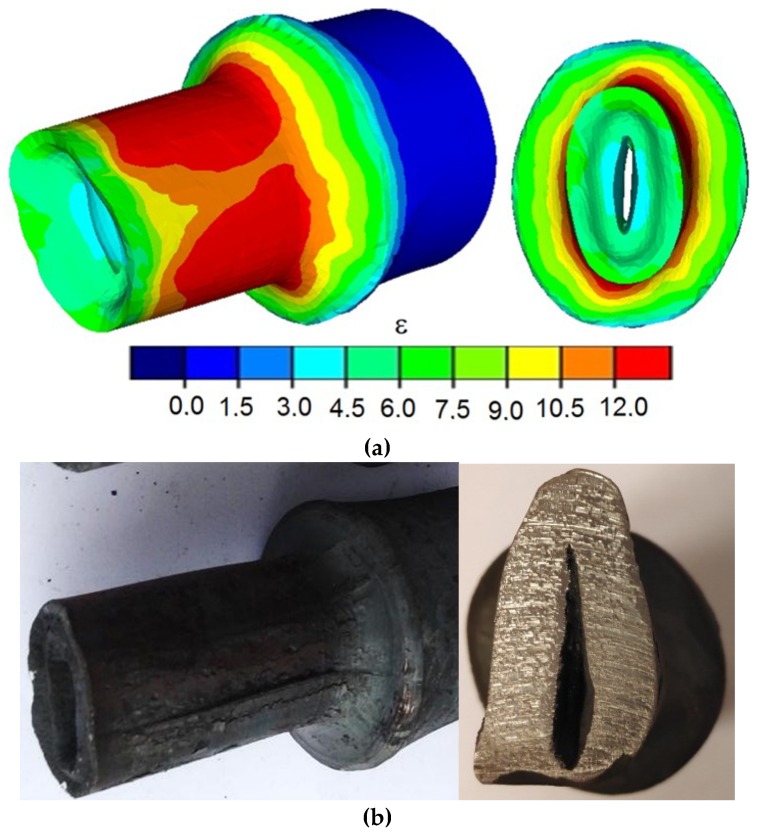
Deformation of the formed step as a result of wall lamination and deformation (process parameters *v/n* = 1 mm/rotation, *t_o_/D* = 0.16, *δ* = 1.5): (**a**) finite elements method (FEM)-determined shape and effective strain distribution; (**b**) shape deformation during laboratory testing.

**Figure 10 materials-12-03049-f010:**
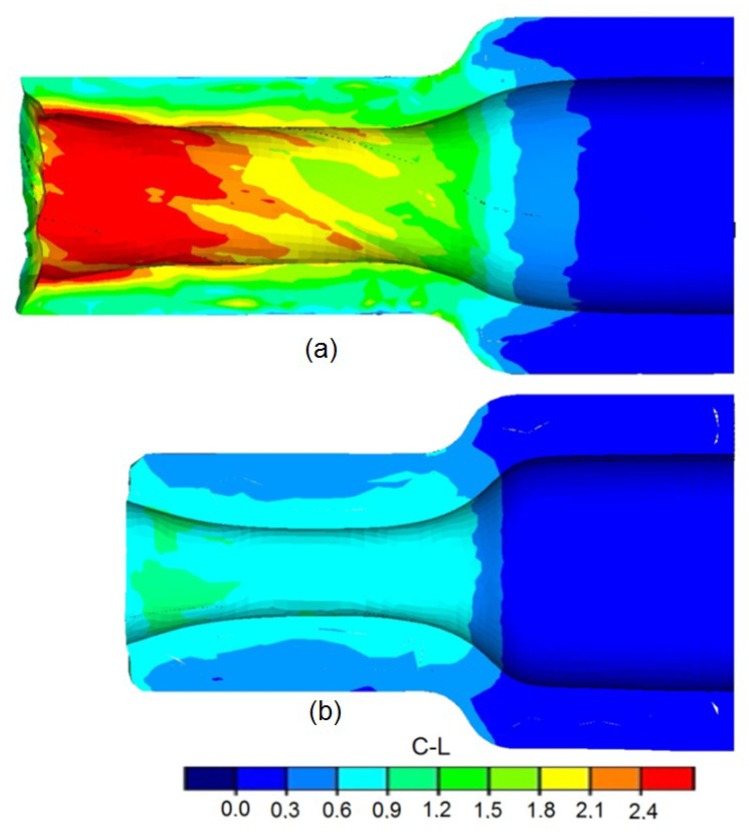
Finite elements method (FEM)-determined distributions of the Cockroft–Latham damage criterion: (**a**) for *δ* = 1.5, *t_o_/D* = 0.16 mm, *v/n* = 2 mm/rotation; (**b**) *δ* = 1.5, *t_o_/D* = 0.16, *v/n* = 6 mm/rotation.

**Figure 11 materials-12-03049-f011:**
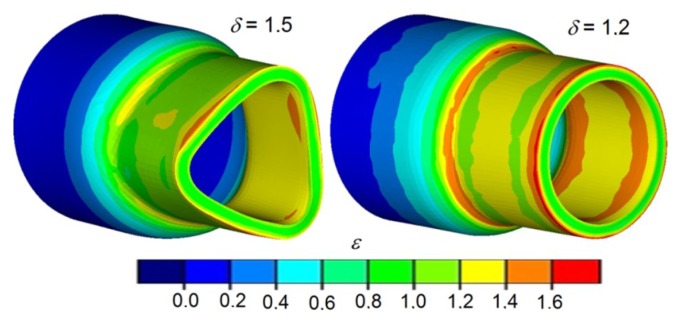
Shape of the forging obtained from thin-walled semi-finished products with small relative wall thickness *t_o_/D* = 0.095 and relative speed *v/n* = 2 mm/rotation.

**Figure 12 materials-12-03049-f012:**
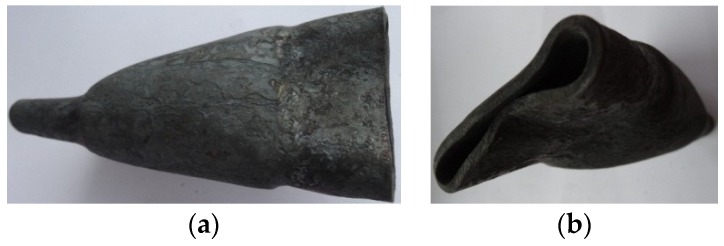
Deformation of the compressed forging, resulting from an overly thin wall of the billet (*v/n* = 4 mm/rotation, *δ* = 1.5): (**a**) front forging view, (**b**) side forging view.

**Figure 13 materials-12-03049-f013:**
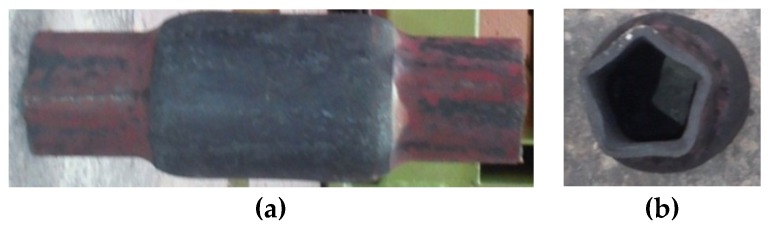
Deformation of the cross-section of the compressed steps, caused by excessive cooling of the material (*v/n* = 0.5 mm/rotation s, *δ* = 1.5): (**a**) front forging view, (**b**) side forging view.

**Figure 14 materials-12-03049-f014:**
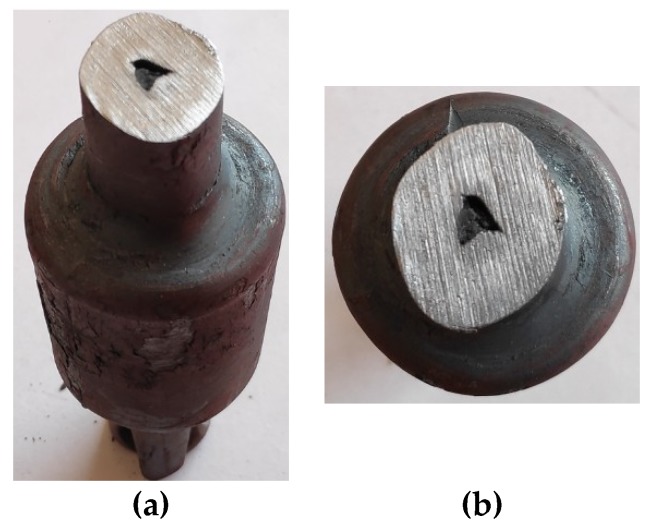
Deformation of the compressed steps of the forging, caused by the gap of the hole closing and the tools slipping (*v/n* = 10 mm/rotation, *δ* = 1.8, *t_o_/D* = 0.16): (**a**) forging view, (**b**) cross section of the forging step.

**Figure 15 materials-12-03049-f015:**
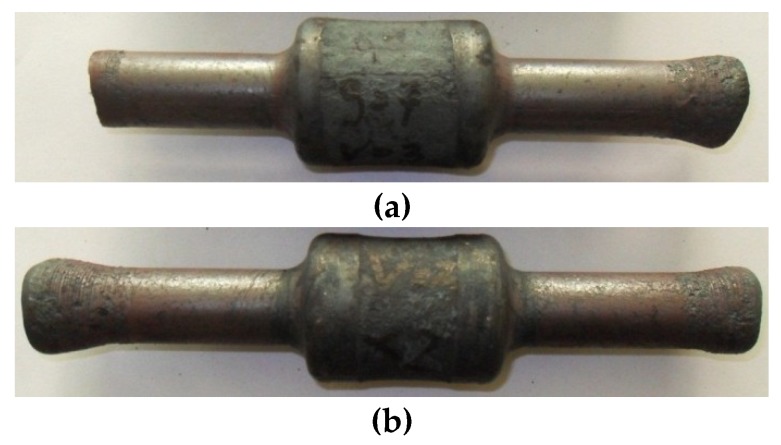
Forgings manufactured in two stages with additional material heating between the operations: (**a**) *t_o_/D* = 0.16; (**b**) *t_o_/D* = 0.29, (*δ* = 1.9, *v/n* = 8 mm/rotation).

**Figure 16 materials-12-03049-f016:**
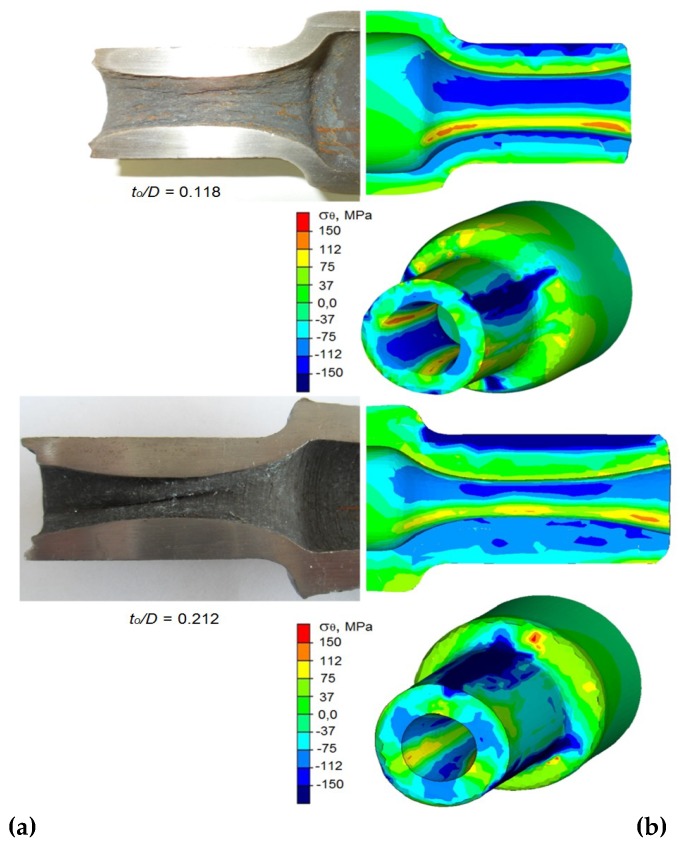
End steps of the forgings with a refraction on the hole surface: (**a**) forgings obtained during the experiment; (**b**) finite elements method (FEM)-determined distributions of the circumferential stresses.

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
