# Peer review of "Limits of the Process of Rotational Compression of Hollow Stepped Shafts"

_materials, 2019, doi:10.3390/ma12183049_

Round 1

Reviewer 1 Report

No major revision required, but will be interesting to apply DOE and statistical analysis of FEM/experimental results comparative

Author Response

Thank you for your comments on our manuscript.

Of course the Reviewer is right about the possibility of applying the DOE techniques in an effective experiment planning, as well as conducting  a static analysis of the  comparative results of FEM and the experiment. Unfortunately, the authors decided against applying advanced statistic methods in their research, instead realising the complex and multi-variant research on rotational compression of hollow elements. Such an approach significantly increased the time of research and the number of tests. It allowed the authors to analyse  a wider spectrum of variants of the combinations of the technological parameters of the process. The statistical analysis of the results was performed only for the parameters ensuring a stable process (allowing one to obtain the correct product). This analysis comprised of strength and energy parameters as well as geometrical parameters of the obtained forgings. Exemplary results of this analysis are presented in another study [2] Tomczak, J. Study of rotary compression process of hollow parts, Lublin University of Technology Press, Lublin 2016. Nevertheless. In the case of industrial implementation of this technology the advanced DAE techniques will be implemented in accordance to the Reviewer’s suggestion.

Reviewer 2 Report

-                     The equation (1) applies to the material C45 and to the temperature in the range 700 - 1250 ° C. Apparently this is for some mean values of this material. The mechanical properties of the material are within a certain tolerance (Re, Rm, ...), do you think the equation will apply to all values? Wouldn't it be appropriate for the model to have a specific mechanical property (Re, Rm, A5 (10))?

-                     Figure 8 d) shows the internal defects that occurred on the forging under certain parameters of forming. In view of the error, I think it would be appropriate to deal with the friction between the material and tool.

-                     From the figures 9b), 13 and 14, it is not possible to clearly identify the surface of the deformed parts that could greatly affect the forming process. 

-                     The contribution is very interesting, it deals with the determination of limit values of hollow shaft rolling parameters. Mastering such technology could bring significant savings in machinery weight. Therefore, I consider the contribution current.

-                     It would be appropriate to make an economic comparison of such production with conventional shaft production.

Author Response

Thank you for your comments on our manuscript.

" The equation (1) applies to the material C45 and to the temperature in the range 700 - 1250 ° C. Apparently this is for some mean values of this material. The mechanical properties of the material are within a certain tolerance (Re, Rm, ...), do you think the equation will apply to all values? Wouldn't it be appropriate for the model to have a specific mechanical property (Re, Rm, A5 (10))?"

An analysis of the limiting phenomena was performer for rotational compression in hot working conditions, assuming the rigid-elastic material model, which does not include the area of the springback. Of course, the flow curves should be determined separately for each case in the plastometric tests. However, in the case of the same material grade the differences in values should not have a significant influence on the value of the plastic strain. During the FEM simulation a model from the program database was used. The authors have verified this model during experimental testing multiple times (also during the processes of rotational forming). Due to this fact, the authors believe that this equation sufficiently describes the behaviour of the material (C45 grade steel) during its forming. As far as the modelling of processes in cold or warm working conditions the flow curves should definitely include the changes to the strength properties of steel, in accordance to the Reviewer’s suggestions.

" Figure 8 d) shows the internal defects that occurred on the forging under certain parameters of forming. In view of the error, I think it would be appropriate to deal with the friction between the material and tool."

Figure 8d presents the inner defects of the formed step, caused by torsion of the cross-section of the end steps of the shaft. This phenomenon is, of course, caused by the friction forces. This deformation might be minimized by decreasing friction. In the majority of these technologies, however, the processes are realised without lubrication, which causes the values of friction forces to be very significant. During the research an attempt at decreasing the values of the friction forces on the surface of material – tool contact was made. However, in this case there is danger of the tools slipping, which leads to deformation and crushing of the formed steps (especially at greater values of tool speed and significant cross-section reductions).

"From the figures 9b), 13 and 14, it is not possible to clearly identify the surface of the deformed parts that could greatly affect the forming process."

In figures 9b and 14 a shape of the cross-section deformed during the process was added.

"It would be appropriate to make an economic comparison of such production with conventional shaft production."

An analysis of the economical profitability of the process was conducted in the early stage of the tecchnology development. Information on these aspects was presented, among others, in the authors’ another study [2] Tomczak, J. Study of rotary compression process of hollow parts, Lublin University of Technology Press, Lublin 2016 and [3] Bartnicki, J.; Pater, Z. Cross-wedge rolling of hollow parts, Lublin University of Technology Press, Lublin 2005. According to the analysis, the rotational compression process allows to decrease the production cost by 40 – 50 %, compared to the traditional methods of metal forming. The profitability of this technology decreases the cost of the billet (tube), which is significantly higher than the cost of semi-finished solid products (rods), used in other methods of forming hollow elements. It is, however, to be stressed that other factors, apart from the cost of the billet, influence the production cost, among others: cost of the tools (tools in the shape of stepped rolls are much cheaper than the tools used in other technologies), insignificant cost of tool regeneration, simple construction of the machines, simple scheme of the process, lower number of operations and lack of cooling lubricants. All of those factors decrease the production cost. It is also important that the above mentioned factors decrease the cost even in the case of small-lot production.